behaviour, evolution, genomics

adaptive radiation, behavioural plasticity, ddRAD, Isla del Coco, translocation experiment, *Wendilgarda galapagensis*

**Author for correspondence:**
Darko D. Cotoras
e-mail: darkocotoras@gmail.com

# Intraspecific niche partition without speciation: individual level web polymorphism within a single island spider population

Darko D. Cotoras[1,2], Miyuki Suenaga[2] and Alexander S. Mikheyev[2,3]

[1]Entomology Department, California Academy of Sciences, 55 Music Concourse Drive, Golden Gate Park, San Francisco, CA 94118, USA
[2]Ecology and Evolution Unit, Okinawa Institute of Science and Technology, Tancha 1919-1, Onna-son, Okinawa 904-0495 Japan
[3]Research School of Biology, Australian National University, Canberra, Australian Capital Territory, Australia

DDC, 0000-0003-1739-5830

Early in the process of adaptive radiation, allopatric disruption of gene flow followed by ecological specialization is key for speciation; but, do adaptive radiations occur on small islands without internal geographical barriers? Island populations sometimes harbour polymorphism in ecological specializations, but its significance remains unclear. On one hand, morphs may correspond to 'cryptic' species. Alternatively, they could result from population, developmental or behavioural plasticity. The spider *Wendilgarda galapagensis* (Araneae, Theridiosomatidae) is endemic to the small Isla del Coco and unique in spinning three different web types, each corresponding to a different microhabitat. We tested whether this variation is associated with 'cryptic' species or intraspecific behavioural plasticity. Despite analysing 36 803 loci across 142 individuals, we found no relationship between web type and population structure, which was only weakly geographically differentiated. The same pattern holds when looking within a sampling site or considering only $F_{st}$ outliers. In line with genetic data, translocation experiments showed that web architecture is plastic within an individual. However, not all transitions between web types are equally probable, indicating the existence of individual preferences. Our data supports the idea that diversification on small islands might occur mainly at the behavioural level producing an intraspecific niche partition without speciation.

## 1. Introduction

Niche partition takes place when different organisms make use of ecological space in their own ways [1]. It often manifests as character displacement owing to competitive exclusion [2–4]. This has been documented *in situ* during decades of field observations, such as in species pairs of Darwin's finches (*Geospiza fortis* Gould, 1837 and *Geospiza magnirostris* Gould, 1837) on the island of Daphne Major in the Galápagos [5]. Niche differentiation plays an important role in the production of new species leading to adaptive radiations [6]. Classic examples include the microhabitat use of Hawaiian *Tetragnatha* Latreille, 1804 spiders [7], feeding strategies of African Cichild fishes [8] and space occupation on Caribbean Anoles lizards [9]. In most of these examples, niche differentiation follows niche expansion after an island colonization event.

Initial stages of niche expansion, prior to niche partition, might require the evolution of phenotypic polymorphism either at the population (different individuals exhibit different phenotypes) or individual levels (the same individual

exhibit different phenotypes/behaviours through its life). How these polymorphisms relate to adaptive radiations remains poorly understood.

The disruption of gene flow followed by ecological specialization is proposed as a major mechanism for the explosive generation of new species in the context of adaptive radiation [10,11]. For example, within the adaptive radiation of the Hawaiian *Tetragnatha* spiders on the middle-aged island of Maui, it is possible to find newly formed species from the same eco-morphology. These species have currently overlapping distributions (secondary contact), however there is no ongoing hybridization [12]. Unless competitive exclusion and extinction of one of those close related species takes place, this situation represents the exact moment prior to ecological differentiation; when reproductive barriers have been already established, yet species are still ecologically equivalent. This supports the idea that some degree of allopatry is required prior to the origins of ecological speciation [10,11]; but, what happens when there is no opportunity for allopatric separation? The prediction that speciation depends on a measure of allopatry can be further tested in cases where ecological specialization has evolved without evident geographical barriers.

One potential test case would be a lineage that has evolved different ecological specializations (i.e. eco-morphologies) while inhabiting a small island. In such a setting, there would have been little opportunity for geographical differentiation, however there is still a potential for niche expansion followed by specialization in order to reduce competition.

A possible scenario to explain the presence of different ecological specializations is that the lineage inhabiting the island may actually be a set of previously non-recognized (cryptic) species. This scenario predicts that each eco-morph would have previously genetically differentiated by some sort of additional processes. For example, by a landscape that seems to be continuous at present, but was fragmented in the past. This is the case for the adaptive radiation of *Miocalles* Pascoe, 1883 weevils on the small island of Rapa, where the mechanism for intra-island speciation was owing to past geographical configurations [13]; or, by a sympatric speciation process, as in the *Howea* Beccari palms on Lord Howe Island, which was driven by the expression plasticity on flowering genes in response to local variations in soil chemistry [14].

An alternative scenario to explain a lineage which has evolved different ecological specializations could be the presence of a single species with high intraspecific variability owing to population, developmental or behavioural plasticity. For example, the Hawaiian happy face spider (*Theridion grallator* Simon, 1900) [15] presents an exuberant colour polymorphism, but their population structure is related to the geography and not a phenotypic variation. Another classic example of population polymorphism is the industrial melanism of the peppered moth, *Biston betularia* Linnaeus, 1758 [16]. In both cases, the origin of the polymorphisms has been related to predatory pressure [17,18], but, several other factors could cause a polymorphic condition, such as niche differentiation [19], balancing selection [20] and standing variation, among others. In these cases, interbreeding between the morphs has prevented speciation.

Genetic tools and field experiments can be used in combination to distinguish between the two scenarios (multiple species versus a single polymorphic species). High-throughput sequencing of independent nuclear markers can detect fine genetic structure, hybridization events and even individual genomic regions under divergent selection (i.e. $F_{st}$ outliers) [21,22]. In particular, the double digest restriction-site associated DNA (ddRAD) approach appears to be ideal for non-model organisms without extensive genetic resources [23]. Complementarily, field experiments allow us to determine the extent of phenotypic plasticity in the population.

## (a) The unique web polymorphism of *Wendilgarda galapagensis* [24]

The Isla del Coco is a small volcanic island (figure 1a) located between the Galápagos (680 km) and the south of Costa Rica (550 km). As other oceanic islands, it is characterized by a depauperate flora and fauna, which has allowed for the niche expansion of the local species owing to ecological release [25,26]. A good example of this is the Darwin's finch from Isla del Coco (*Pinaroloxias inornata* Gould, 1843). It presents a diversity of foraging behaviours equivalent to what is expected to be seen across many different families of birds. As a species it is a generalist, however, it is composed by year-around specialized individuals on nine different feeding strategies [19], showing a striking example of behavioural diversification within a single species, which results in a within-population niche differentiation [19]. Similarly, the endemic spider *Wendilgarda galapagensis* [24] appears to present an extreme level of intraspecific variation with three different types of webs [27].

The genus *Wendilgarda* (Araneae, Theridiosomatidae) is composed of 14 named species with tropical distribution [28,29]. Altogether, this group presents highly modified webs not present in any other spider species [30]. It is believed that the ancestral web of *Wendilgarda* is composed of approximately six horizontal non-sticky suspension lines (homologous to the radii of an orb-web) located 1–4 cm over the water, which radiate from a central area; and several (1–16 per suspension line) vertical sticky lines (homologous to the sticky spirals) that come into contact with the water [28,31,32]. Its sister genus, *Epilineutes* [28], spins a type of orb-web [31].

In *W. galapagensis*, the basic web design of the genus is expanded into a total of three different types with respect to its mainland counterparts, suggesting occupation of additional ecological niches [33–35].

The first type is the water web, which is the same as in the other *Wendilgarda* species (figure 1b). The low land (land) web (figure 1c) is similar to the water web, but it is found over dry land and the vertical lines are longer. Regardless of the structural similarity, the sequence of addition of vertical lines by the spider in both cases is very different [27,36]. Finally, the high land (aerial) web (figure 1d) does not have contact with water or the ground, and it consists of a series of horizontal lines connected to branches, rocks and/or leaves. In this case, the sticky lines are fewer and longer, they radiate in different directions from the central area. There is also no clear pattern in the order of addition of sticky lines on aerial webs [27]. The differences between the three web types go beyond changes in the number of repetitions of a given architectural element or relative sizes, as is the case for the web plasticity reported on cobweb [37–39] or orb-web spiders [40–42].

These three types of webs occur, literally, right next to each other in the field, while still placed on their respective microhabitats (electronic supplementary material, figure S1). Individuals presenting the different web types occupy different niches related to microhabitat and food sources available

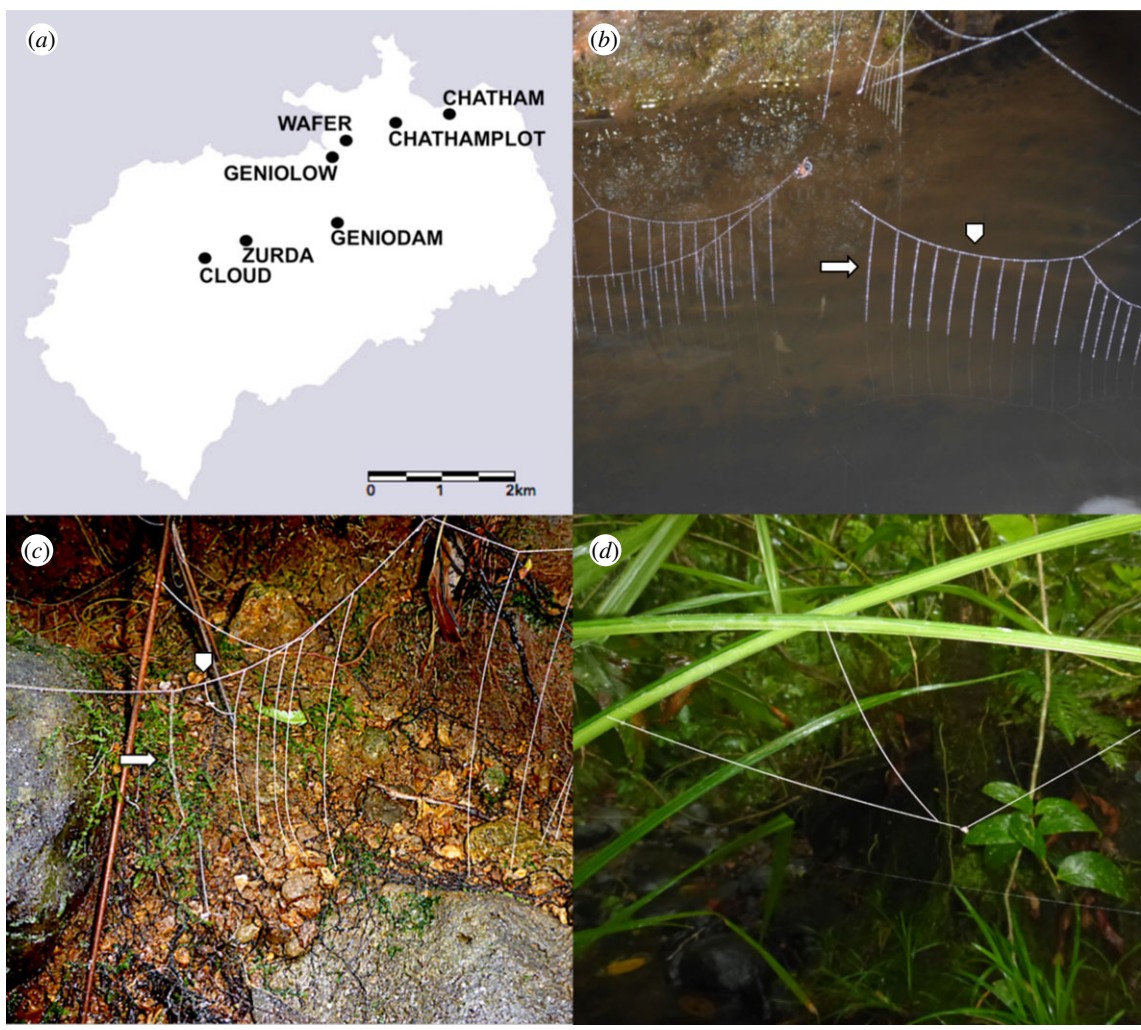

**Figure 1.** Study site and web types. (*a*) Map of Isla del Coco with collecting sites (map downloaded from www.esri.com), (*b*) water web, (*c*) low land (land) web and (*d*) high land (aerial) web. The arrows indicate vertical sticky lines and arrow heads the horizontal non-sticky suspension lines, respectively. Silk lines on (*c*) and (*d*) have been re-traced to make them more evident. Original pictures in the electronic supplementary material, file S1.

therein. Each web type is microhabitat-specific, exposing the spider to different environmental conditions (wind, humidity, exposure to predators, probability of being destroyed by an overflow, etc.). Also, each microhabitat provides different potential prey species. The spiders with water webs could capture insects hovering or walking over water, which has been reported for mainland *Wendilgarda* [32]. Ones with high land webs are restricted to winged insects. In the case of the low land web, their most likely prey is walking insects. In other words, the web types spun by each spider expose it to a different niche [43].

The three web types of *W. galapagensis* differ in the general structure, microhabitat placement and behaviour associated with its constructions [36], making it an exceptional case of web polymorphism. This diversification on the web design is remarkable considering the high degree of conservatism on web types, in many cases being a diagnostic character at the family level [44–46].

The extent of individual plasticity in web-building behaviour on this species is not well known. Transitions between web types have been only reported as a single incidental field observation, where individuals from a low land web transformed it into a water web after a flooding event [27]. However, there is no information about changes between the other web types or how often they occur. Furthermore, even if the web architecture is somewhat plastic, it may still

have underlying genetic predispositions that could lead to eventual differentiation.

Within the adaptive radiation of the Hawaiian *Tetragnatha* web-building clade, it has been shown that differences in web architectural traits and microhabitat selection are associated with differential trophic niches allowing for the co-occurrence of sympatric species [47–49]. In the case of *W. galapagensis*, the three web types are associated with different microhabitats (over water, dry land and vegetation), which is reminiscent of niche partitioning in well-studied adaptive radiations [7,9]. Given that niche differentiation in the context of an adaptive radiation is associated with different species, and it is known that spider webs can present architectural plasticity, here we set out to ask: is the variation in *W. galapagensis* web types associated with different species—not yet recognized—, or does it represent an exceptional intraspecific behavioural plasticity?

If genetic differentiation is defined by web type, this would suggest an ongoing or past ecological speciation event implying that *W. galapagensis* is really a species complex, which has possibly undergone adaptive radiation. Therefore, experimentally translocated individuals between microhabitats should not be able to spin a different type of web. Note the genetic signature of this alternative could be detectable by the presence of $F_{st}$ outliers corresponding to specific regions under selection associated with each web type. If this were the case, it will

correspond to one of the few examples of *in situ* speciation on a small island. On the other hand, if there is a lack of population structure defined by web type, then experimentally translocated individuals between microhabitats should be able to spin a different type of web according to their new placement, implying that the web variation was produced by intraspecific behavioural plasticity.

In order to answer this question, we combined population genetics with field experiments. First, we collected individuals of *W. galapagensis* from the three web types from different localities on the island, and then, prepared ddRAD libraries [50] to assess the potential population structure associated with the web type. Second, we performed field translocation experiments, whereby we used mark-recapture to test if an individual was able to change its type of web after being moved to a different microhabitat. This data allowed testing of whether unparalleled polymorphism of *W. galapagensis* web types is the result of a speciation process, or the expression of behavioural plasticity.

## 2. Methods

### (a) Study site

The Isla del Coco (05°31'4.79" N, 87°04'10.80" W) is located 550 km off the coast of Costa Rica and 680 km from the Galápagos archipelago. It is a small volcanic island (24 km$^2$) originated at the Galápagos hotspot. The age of the currently exposed land ranges between 1.9 and 2.4 Myr. The highest elevation is Cerro Iglesias (634 m). A high annual precipitation (7 m) sustains a large tropical rainforest [51]. Most of the forest on the island is classified as premontane rainforest, while on the highest elevations it has cloud forest. It is remarkable that the cloud forest appears as low as 450 m, which is only possible owing to a nearly constant cloud cover [52]. The logistic difficulties associated with the condition of isolation make the access to the island extremely challenging, therefore invertebrates have been poorly surveyed [53–56]. Indeed, the last published arachnid survey reported a total of only 50 species representing 26 families in six orders [57].

### (b) Population structure

#### (i) Field collection

Isla del Coco was visited between the 8 and 25 July 2017. Day and night surveys were performed along small creeks and streams (electronic supplementary material, table S1; figure 1*a*). Spiders were detected by dusting flour over the webs. A total of 142 individuals were collected from seven sampling sites: 54 water webs, 60 high land webs and 28 low land webs. Once identified the webs were photographed and specimens collected for future preservation in 100% ethanol.

#### (ii) DNA extraction and library preparation

Full-body DNA extraction was performed following the method described by Tin *et al.* [50]. Double digestion (EcoRI and MseI) and genomic library preparation were performed based on Tin *et al.* [58].

Genomic libraries were sequenced on two lanes of 50 cycles single-end reads on a HiSeq2500 (Illumina, Inc.) at the DNA sequencing section of the Okinawa Institute of Science and Technology.

### (c) Data processing

Adaptors and polymerase chain reaction duplicates were removed following Tin *et al.* [58]. Raw and adaptor trimmed

sequences for 142 libraries were used as input to run the dDocent pipeline [59,60]. We used VCFTOOLS v. 0.1.12b [61].

For more details on DNA extraction, library preparation and data processing see the Methods section in the electronic supplementary material.

### (d) Data analysis

#### (i) Marker discovery

The numbers of raw and filtered single nucleotide polymorphisms (SNPs) were obtained from the vcf files generated by FREEBAYES v. 1.1.0 [62] and VCFTOOLS v. 0.1.12b, respectively. The estimation was done with the function stats from BCFTOOLS v. 1.3.1 [63]. The number of biallelic SNPs used for downstream analysis were obtained after the transformation of the filtered vcf file into a Genlight format with the R package vcfR [64].

#### (ii) Principal component analysis

We performed a principal component analysis (PCA) looking at all the specimens together and separating them by collecting site. When all samples were considered, we labelled samples by web type or collection site, separately. While on the per site PCA, we only labelled the samples by their web type.

The PCA was done with the function glPCA (R package adegenet [65,66]) using only the biallelic sites on Genlight format. The plot was done with *ggplot* (R package GGPLOT2) [67].

#### (iii) Pairwise $F_{st}$ estimation

All potential pairwise $F_{st}$ comparisons (Weir and Cockerham mean $F_{st}$ estimate' and 'Weir and Cockerham weighted $F_{st}$ estimate' [68]) were calculated with VCFTOOLS v. 0.1.12b.

#### (iv) $F_{st}$ outliers

Because it is possible that neutral genome-wide differences do not correspond to ecological differentiation between incipient species, we focused on a more detailed analysis using $F_{st}$ outliers, which could potentially be targets of recent selection. They were selected with the R package pcadapt [69,70]. We examined whether there was significant $F_{st}$ differentiation (computed as above) between the web types using paired *t*-tests.

#### (v) Mantel's test

In order to test isolation by distance, a Mantel test with 9999 repetitions was performed using the R package ade4 [71]. We tested the correlation between the $F_{st}$ matrix (electronic supplementary material, table S3) and the linear distance between collecting sites (electronic supplementary material, table S5).

### (e) Translocation experiment

#### (i) Experimental design

The goal of the experiment (23 January to 2 February 2019) was to test if an individual of a given web type can spin a different type of web after translocation to a new microhabitat.

Two different treatments were performed on each site: (i) control: spiders originally from one microhabitat were marked and returned to the same microhabitat; and (ii) translocation: spiders originally from one microhabitat were marked and transferred to a different microhabitat. Specific site information is in the electronic supplementary material, table S2 and pictures of each site in the electronic supplementary material, figure S2. Because low land webs are very uncommon the experiment was focused on high land (aerial) and water webs.

On average 18.2 ± 6.2 individuals per site were marked using different colours of nail polish (electronic supplementary material, figures S3 and S4; file S2). The colour represented the

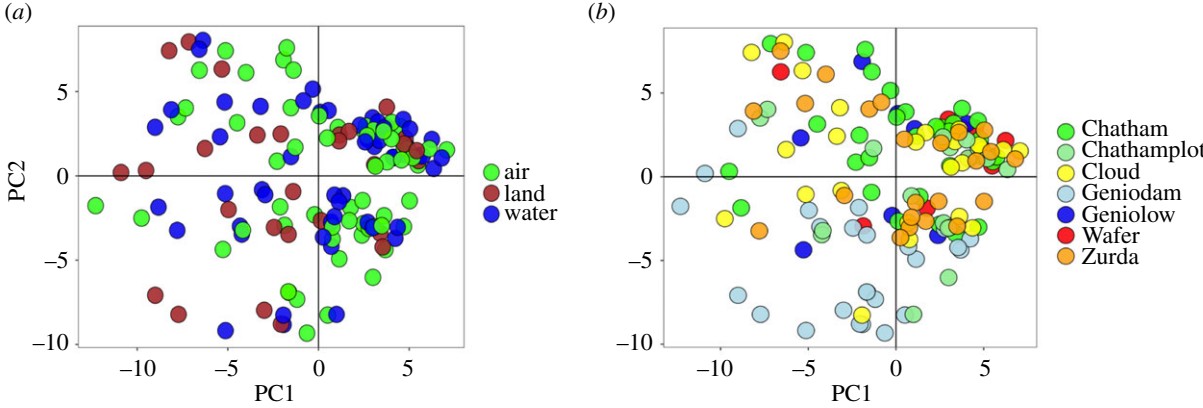

**Figure 2.** Principal component (PC) analysis of all the specimens. (*a*) Specimens labelled by web type, (*b*) specimens labelled by collecting site. (PC1 = 2.09%; PC2 = 1.73%).

web type where the individual was originally found: water web, white; high land (aerial) web, green; and low land (land) web, purple.

After marking and reintroduction the spiders were monitored up to five times during a period of 2 days (electronic supplementary material, table S2). On each observation round, re-collected spiders were scored based on their microhabitat (air or water) and the type of web (aerial or water). For more details, see the Methods section in the electronic supplementary material.

## (f) Data analysis

Transition probabilities between the original web type and the one recorded after experimental treatment (control or translocation) were calculated based on the total number of recaptures on a given observation (I–V) from the same treatment. Differences between control and translocation were tested with a Fisher's exact test for count data implemented in the function *fisher.test* from the R package stats v. 3.6.1.

## 3. Results

## (a) Population structure

### (i) Marker discovery

For assembly purposes, first we grouped all specimens as if they belonged to a single population. The total number of raw and filtered reads was 130 930 and 38 375, respectively. The filter allowed only SNPs present in at least 90% of individuals. When, considering each sampling site as a different population a total of 131 371 raw SNPs were discovered and 44 716 retained after filtering. The last grouping scheme was selected for downstream analysis. It yielded 36 803 biallelic SNPs in a total assembly size of 6.5 Mb.

### (ii) Principal component analysis

The PCA of all the specimens together shows a homogeneous distribution of samples on the PC space regardless of web type (figure 2*a*) or sampling site (figure 2*b*). There is a slightly denser concentration of specimens in the upper right corner of the chart, but it does not correlate with any of the tested variables.

The per site PCA also does not show evident grouping by web type (electronic supplementary material, figure S5). Three sites do not have specimens with a land web type either because they were not found or the libraries failed.

### (iii) Pairwise $F_{st}$ estimation

All the possible comparisons present values of less than 0.01 for the Weir and Cockerham weighted $F_{st}$ estimate (electronic supplementary material, table S3). In general, the highest $F_{st}$ values are when comparing with the Cloud site. Within this context, the pairwise comparison Zurda and Wafer also present high divergence values. The same tendencies are valid for the regular Weir and Cockerham $F_{st}$ estimate (electronic supplementary material, table S4).

### (iv) $F_{st}$ outliers

A total of 1418 SNPs were selected as $F_{st}$ outliers. When partitioning by web types, none of the groups showed significant $F_{st}$ differentiation (paired $t$-test, $p > 0.05$ in all cases).

### (v) Mantel's test

The observed correlation between the $F_{st}$ and the geographical distance matrixes was positive ($r = 0.65$) and significant ($p = 0.0025$), implying the presence of isolation by distance.

## (b) Translocation experiment

Translocation and control individuals could change web types (electronic supplementary material, figure S6). However, transition probabilities between web types were unequal.

During data collection (electronic supplementary material, file S2), it was noticed that aerial webs could be sub-divided in two different kinds: (i) 'aerial web in the air', consisting of the typical aerial web, and (ii) 'aerial web over water', which has the same morphology as the typical aerial web, but it is placed right above the water, at a distance that other spiders would spin a water web (electronic supplementary material, figure S7).

Based on the first (I) observation, 4.6 hrs. ± 53 min after the beginning of the experiment (recapture = 29.6%) (figure 3), there are no significant differences on the total counts of the transitions between web types, both in control and translocation treatments, for neither the individuals which originally were on a water web ($p = 0.646$) nor those from an aerial web ($p = 0.058$). There were also no significant differences when aggregating both treatments (control + translocation) and comparing the response of individuals which were originally on a water web or an aerial web ($p = 0.866$).

Individuals which were originally on a water web (figure 3*a*), in either control or translocation treatments, most

**Figure 3.** Transition probabilities between web types after experimental treatment. Individuals originally found on (*a*) water web and (*b*) aerial web. The shape and colour of the boxes represents the microhabitat: green rectangle, aerial; grey circle, water. The text inside the box refers to web type. The original web type presented on the two boxes on the sides of the diagram and the produced web type on the three boxes on the centre of it. The percentage over the arrows represents the transition probabilities with respect to all the recaptured individuals on the same treatment. The numbers in parenthesis correspond to the actual counts.

frequently transitioned to an aerial web in the air, followed by an aerial web over water. Construction of the water web was the least common, with only one occurrence in the control and no occurrences in the translocation treatment.

For the individuals which were originally on an aerial web (figure 3*b*), in the control treatment the most common response transition was to spin an aerial web in the air, followed by an aerial web over water and then a water web. In the translocation treatment, the most common web was aerial over water, followed by an aerial in the air. No spider produced a water web after translocation.

For the later observations II–V (electronic supplementary material, figure S8 and table S6), the same general patterns are observed (see the Results section in the electronic supplementary material).

Two individuals from a low land (land) web were marked from sites 7 and 9 (both Control treatments). The one from site 9 was found on observation II spinning an aerial web in the air (electronic supplementary material, figure S6d).

## 4. Discussion

The process of adaptive radiation requires an early phase of allopatric speciation in order to establish reproductive barriers, which will allow for ecological differentiation [10,11]. This initial disruption of gene flow is possible in areas with topographic or habitat complexity, which is not always the case on small and isolated islands. We tested whether an ecological adaptation (i.e. web architecture) was associated with incipient speciation in a spider found on that kind of island to understand the role of geographical isolation at the beginning of adaptive radiation.

Remarkably, despite its unparalleled diversity of web types associated with microhabitats, we detected no genetic structure associated with this trait in *W. galapagensis* (figure 2*a*). The same pattern holds when testing genetic structure by web type on each site individually (electronic supplementary material, figure S5) or looking only at $F_{st}$ outliers.

The lack of genetic structure defined by web type is supported by the translocation experiments, which showed that the same individual can spin different web types (electronic supplementary material, figure S6). Therefore, web architecture is an individually plastic trait rather than a population- or species-specific trait. Despite this individual level plasticity, not all the transition probabilities are equal (figure 3; electronic supplementary material, figure S8). There is a

tendency for spiders from both web types to spin an aerial web (in the air or over water) after the experimental perturbation, in both the translocation or control treatments.

This preference for the aerial web, which represents an evolutionary innovation exclusive to *W. galapagensis*, could be explained in two ways. First, the number of available spots for a water web is more limited than the ones for an aerial web (electronic supplementary material, figure S7). Water webs are restricted to the surface of the water in the areas with attaching points and a weak current. By contrast, aerial webs may be placed in more locations owing to the non-surface dependent nature of their architecture. A second alternative explanation is that the experimental manipulation itself triggers an escape behaviour from the water surface. On Isla del Coco, sudden strong rains could increase water levels, potentially threatening spiders located next to the water surface. Finally, web architecture and behavioural biases could have genetic components that may provide the raw material for eventual speciation under the right conditions of gene flow and selection. Regardless of the mechanism, all these lines of evidence suggest that niche occupation owing to web type is not strictly defined for an individual, instead it can dynamically change. Similarly, web types are not associated with the presence of different species.

Isla del Coco is small, with the most distant populations located only 3.83 km apart and at about 500 m of altitudinal difference. They have only a low level of genetic differentiation ($F_{st} = 0.009$). Nonetheless, we could detect significant isolation by distance suggesting that our analysis detected subtle differences in structure between populations. However, different web types could be found, literally next to each other, and spiders can switch between them, minimizing the potential for parapatric speciation.

What is the origin of observed intraspecific niche differentiation of *W. galapagensis*? The argument of competitive exclusion as a driving force of adaptive radiations applies differently to *W. galapagensis*. The different niches, characterized by web types, are dynamically used by individuals from each web type, instead of being associated with an individual species. As shown by the translocation experiment, individuals are able to switch from one web type to the other.

As in adaptive radiation, this intraspecific niche partition may reduce competition. In addition, it could be explained alongside different elements present in models commonly used to understand web architectural plasticity (reviewed by [72]). For example, plastic extended phenotype models suggest spiders have a dynamic adaptation to given ecological

circumstances. Similarly, optimal performance models argue for adaptation to local prey and environmental conditions. Finally, optimal foraging models indicate that there is optimization on the energy investment required to obtain resources.

In contrast with adaptive radiations, this intraspecific niche partitioning is not linked with speciation events. While in other archipelagoes niche partition is often associated with explosive speciation events, on Isla del Coco, it remained as a species-level trait. One of the biggest differences with other spider adaptive radiations is the fact that *W. galapagensis* is confined to a single small island. By contrast, the adaptive radiation of the Hawaiian *Tetragnatha* spiders [7] and the *Dysdera* Walckenaer, 1802 spiders from the Islas Canarias [73] occur across whole archipelagos. On those island systems, a single volcano would be bigger than the whole Isla del Coco.

Previous studies have explored the idea of a minimum island size for *in situ* speciation, which appears to be taxon dependent and related to dispersal abilities [74]. Taxa with a large degree of gene flow will require a larger minimum area for speciation. As a consequence, their speciation probability will be lower in any given area [74]. This is also consistent with the idea that even sympatric speciation requires some degree of allopatry during its early stages [10,11].

Of course, actual physical allopatry is not strictly necessary and speciation can occur via other mechanisms of reproductive isolation. For example, *Rhagoletis* (Loew, 1862) flies from eastern North America are a classic example of rapid sympatric speciation. Although speciation happened within overlapping geographical ranges, the phenology of the respective hosts was different, separating temporally the reproductive events [75]. Another example is the sympatric species of *Howea* palms from Lord Howe Island, which speciated based on changes in the flowering time induced by local soil conditions [76,77]. However, it is not clear that mechanisms creating this sort of reproductive isolation exist in *W. galapagensis*. Unlike flowering phenology, webs are always present and there is no evidence of males actively discriminating between web types.

An outstanding question is whether the intraspecific niche differentiation of *W. galapagensis* represents a rare case. The Darwin's finch from Isla del Coco (*P. inornata*) has also diversified behaviourally, yet not speciated or even developed morphological differentiation [19,78]. Each individual could present one of nine different year-around stable foraging behaviours, a diversity of behaviours not seen in any other single Galápagos finch species [19]. Although there are no comparable genetic studies to show its genetic homogeneity, systematists refer to this species as a single biological unit [79].

*W. galapagensis* and *P. inornata* present extreme behavioural diversification owing to discrete niche occupancies (foraging strategies) without species formation. In the absence of the possibility of alloparty, these species remain as single biological units, producing an 'intraspecific adaptive radiation' [80], which could alternatively be called 'behavioural radiation'. Regarding the intraspecific niche partition [81,82], also referred as inter-individual or individual niche variation [83,84], the niche limits might not be strict, but rather dynamic instead. Within the total species niche, at any given time, single individuals might occupy a fraction of it preventing intraspecific competition [85]. However, with the restriction of differential transition probabilities (i.e. preferences), they might also be able to use another fraction of the species niche in another moment. Therefore, in an intraspecific adaptive radiation, the niche partition is not as strict as what happens between species in an adaptive radiation.

Fitting this description, two additional examples from cone snails could be classified as intraspecific adaptive radiations. First, is the Rapa Nui endemic *Conus miliaris pascuensis* (Conidae) [26,86–88]. It also has an expanded trophic niche and bathymetric distribution that is explained by the absence of other *Conus* species (ecological release) [25,26] and the appearance of local genetic variants associated with its venom [89]. This sub-species, established 0.45 Ma, also does not have population structure within the island [88].

An analogous situation can be found in *Conus californicus*, a single species present along the California coast with isolation as old as the upper Miocene (*ca* 7 Ma, Stanton, 1966). Similarly, this species has a wider trophic niche with respect to tropical congeners [90,91], and its niche expansion has been associated with novel venom components [92].

Both *Conus*, *W. galapagensis* and *P. inornata* represent isolated lineages from a clade, which have expanded their niches owing to a change in foraging behaviour. For the spider and the finch, there are explicit characterizations for the individual foraging strategies. One might predict the same pattern may be elucidated with further studies on the *Conus* species.

In summary, niche expansion appears to be a common pattern after island colonization [34,93]. It is facilitated by ecological release owing to the depauperate biodiversity condition of the isolated territory [25,26]. It can result in adaptive radiations on islands with complex topology, where some degree of allopatric separation is possible in the initial stages of diversification. Here, the niche expansion occurs at the level of the whole clade and each individual species specializes in a partition of it. By contrast, on small islands without the opportunity for allopatry, it results in behavioural or intraspecific adaptive radiations (a single species evolves a wide niche occupation with an intraspecific niche partition), where the niche expansion remains as a species-level property. The generality of intraspecific adaptive radiations occurring on small islands needs to be further tested with more organisms and other geographical locations, ideally making comparisons of niche occupancy within the same species group, where some members have radiated in an archipelago, while others are only present on small isolated locations.

**Ethics.** The field collection was done under the Comisión Nacional para la Gestión de la Biodiversidad (CONAGEBIO, Costa Rica) permits numbers R-032-2017-OT-CONAGEBIO, R-051-2018-TCONAGEBIO and R-059-2018-OT-CONAGEBIO. The field experiment was done under the Ministerio de Ambiente y Energía – Sistema Nacional de Áreas de Conservación – Área de Conservación Marina Cocos (MINAE-SINAC-ACMC, Costa Rica) permit number 18-2018-I-ACMC.

**Data accessibility.** The genetic data generated in this publication consist of single-end reads produced on two lanes of 50 cycles on a Hiseq2500 at the DNA sequencing section of the Okinawa Institute of Science and Technology. Raw data is deposited at the Sequence Read Archive (SRA; https://www.ncbi.nlm.nih.gov/sra) under the BioProject accession number PRJNA656103. The raw counts from the translocation experiments are on Supp_File_2.xlsx.

**Authors' contributions.** D.D.C. designed the study, secured partial funding, planned and performed field collections, planned and performed field experiments, analysed the data, wrote the manuscript and revised final version. M.S. performed laboratory work and revised the final version. A.S.M. secured partial funding, analysed the data, wrote the manuscript and revised the final version.

**Competing interests.** We declare we have no competing interests.

**Funding.** This publication was funded by the Percy Sladen Memorial Fund (The Linnean Society of London), the Herb Levi Memorial Fund for Arachnological Research (American Arachnological Society), the Okinawa Institute of Science and Technology Graduate University and personal resources from D.D.C.

**Acknowledgements.** We are grateful to a lot of people who helped us to complete this research. Particularly to William Eberhard for insightful conversations and suggestions. We are grateful to Rafael Bonilla, Carina Canosa, Juan Carlos Castanedo, Carlos García Robledo, Christopher Grinter, Jean Carlos Martínez, Daniel Soto, Darrell Ubick, Juan Carlos Villalobos and Carlos Víquez; Alberto Hernández Quintero from Servicio Nacional de Guardacostas; Gary Araya, Rolando Cordero, Alonso Esquivel and Michael Moncrieffe from Cuerpo de Bomberos de Costa Rica; Victor Acuña, Eduardo Alvarado, Filander Ávila, Guillermo Blanco, Marta Bogantes, Lucas Campos, Isaac Chinchilla, Geiner Golfin, Moisés Gómez, Esteban Herrera and Sam Medina from Parque Nacional Isla del Coco; and Melania Muñoz from Comisión Nacional para la Gestión de la Biodiversidad (CONAGEBIO) of Costa Rica, for all their assistance and support during fieldwork. We are also grateful to Qiu Lijun and Maéva Techer for support during laboratory work and data analysis, respectively; and Saori Chappell for help with general logistics. We appreciate the discussions with Davor Cotoras and Pabla Viedma, as well as the suggestions from anonymous reviewers.

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
