## [Peer Review File · Proceedings of the Royal Society B: Biological Sciences]

Review History

RSPB-2020-2045.R0 (Original submission)

Review form: Reviewer 1

Recommendation

Accept with minor revision (please list in comments)

Scientific importance: Is the manuscript an original and important contribution to its field?

Good

General interest: Is the paper of sufficient general interest?

Good

Quality of the paper: Is the overall quality of the paper suitable?

Good

Is the length of the paper justified?

Yes

Should the paper be seen by a specialist statistical reviewer?

No

Do you have any concerns about statistical analyses in this paper? If so, please specify them explicitly in your report.

No

It is a condition of publication that authors make their supporting data, code and materials available - either as supplementary material or hosted in an external repository. Please rate, if applicable, the supporting data on the following criteria.

Is it accessible?

N/A

Is it clear?

N/A

Is it adequate?

N/A

Do you have any ethical concerns with this paper?

No

Comments to the Author

Review of manuscript RSPB-2020-2045

The present manuscript by Cotoras et al. reports the results from a population genetics study of a species of spider endemic to a small oceanic island that exhibits a remarkable diversity of web morphotypes. By comparing SNP distribution via RADseq between spiders collected from different web phenotypes it was tested, whether the different web morphotypes indicate cryptic species or if this is a case of phenotypic plasticity. The results clearly support the latter hypothesis.

Overall the study appears well conducted – especially the number of samples and loci is quite impressive and probably represents the largest assay of that kind so far for a population of spiders. The writing is very clear, although there are some typos and errors in the language, so the text needs a thorough check. The figures are appropriate.

This is clearly an interesting study and a remarkable example of phenotypic plasticity, but not more.

I am a bit sceptical about the author's discussion and interpretations and strongly suggest a revision. Especially the caveats of the approach must be more clearly communicated.

First, I am not sure if one may speak of 'niche partitioning' without speciation. Maybe the problem is that it remains unclear, what the authors mean by 'niche' in this context. The authors show that there is no correlation between population structure and web morphotype and interpret web diversity as the product of behavioural (phenotypic) plasticity. This means that each individual is capable to produce any of the three web morphotypes depending on the microhabitat structure. The 'niche' concept is usually applied to species, and thus in this case it simply means the species has an enormous niche width, but not that the niche is partitioned in any way between species, not even populations.

Then, it remains unclear if/how different web building behaviour correlates with gene variation and whether the observed phenomenon represents intra- or inter-individual plasticity. It may still be that there are genes correlating with the expression of web building behaviour that were not caught with the chosen RADseq approach. Ideally, the genomic approach would be combined with a behavioural experimental approach, which is easily implemented: Transplanting individuals into different microhabitats and observing the type of web built. For the initial question posed by the authors, it would be very important to clarify this aspect.

Another aspect the authors highlight repeatedly is that this case of web building plasticity is 'outstanding'. But what does that mean. To clarify this the findings should be better embedded into the state of art knowledge on spider web plasticity, as there is a wealth of information on this

topic.

So this leaves the question about the originality of these findings. The major finding is that the different web morphotypes are all built by the same species, but this has already been proposed by Bill Eberhard in 1989 (“Niche expansion in the spider *Wendilgarda galapagensis* (Araneae, Theridiosomatidae) on Cocos Island”).

Further specific comments below.

Data Statement:

I could not find the Short Read Archive (SRA). Is the NCBI Sequence Read Archive (SRA) meant? An URL should be given.

Title:

I have a problem with the term ‘niche partitioning’ in this context, as the observed phenomenon clearly is plasticity and no evolutionary process is demonstrated. Also see comment above. I suggest to change or omit the term.

Abstract

l. 39: What is meant with ‘population plasticity’? Polymorphism? Also, I would add developmental plasticity here.

l. 45f: This is not a complete sentence. Maybe “This implies that...”.

Introduction

l. 57. word is missing here, e.g. “make use of”

l. 122: typo “Galápagos”

l. 136: the radii of an orb web

ll. 140-154: One or three sentences should be added on the function of these web morphotypes, as it is important for the understanding of niche partitioning.

l. 157: typo “a individual”

l. 159: “how often they occur”

l. 162: “web-building” ?

Material and Methods

l. 197: omit “ago” (either the authors refer to the age which is measured in million years or to the origin which is given in million years ago)

Discussion

In the discussion, some aspects are missing. For instance, there is a bulk of information on phenotypic plasticity in spider webs (e.g. reviewed by Boutry, C., & Blamires, S. J. (2013). Plasticity in spider webs and silk: an overview of current evidence. *Spiders: Morphology, Behavior and Geographic Distribution*, 1-46. and Blamires, S. J., Zhang, S., & Tso, I. M. (2017). Webs: diversity, structure and function. In *Behaviour and ecology of spiders*. pp. 137-164. Springer, Cham.)

Also, I would briefly discuss the possibility if the behavioural plasticity may eventually lead to genetic change and speciation according to the theory by West-Eberhard, or if the results rather oppose to that idea.

l. 339. see my comments above on niche and niche partitioning.

l. 344: The other examples given are all much larger species. *Wendilgarda* are extremely small spiders, so it all depends on their dispersal abilities if the island is really ‘so small’ for them. Is there anything known on the mobility and dispersal abilities of these spiders (e.g. their locomotion radius and if they are able to balloon)?

l. 346f: It is unclear what is meant with this sentence. Needs to be rephrased.

l. 361: So this seems to be an example where environmentally induced developmental plasticity preceded genetic change. This should be discussed.

Figures

Fig. 1: In the web photos it would be good to mark or highlight the sticky lines, so that the reader

may better understand the homologous parts and web function. In d, the web structure remains unclear. It looks like not all silk lines were enhanced. At least one additional radius is missing, otherwise the web would collapse. Also, there seem to be additional silk lines underneath, which were not highlighted. Also it is unclear, where the sticky lines are in this picture.

Review form: Reviewer 2

Recommendation

Major revision is needed (please make suggestions in comments)

Scientific importance: Is the manuscript an original and important contribution to its field?

Acceptable

General interest: Is the paper of sufficient general interest?

Marginal

Quality of the paper: Is the overall quality of the paper suitable?

Good

Is the length of the paper justified?

Yes

Should the paper be seen by a specialist statistical reviewer?

No

Do you have any concerns about statistical analyses in this paper? If so, please specify them explicitly in your report.

No

It is a condition of publication that authors make their supporting data, code and materials available - either as supplementary material or hosted in an external repository. Please rate, if applicable, the supporting data on the following criteria.

Is it accessible?

Yes

Is it clear?

Yes

Is it adequate?

Yes

Do you have any ethical concerns with this paper?

No

Comments to the Author

Cotoras and co-authors use a tiny web building spider endemic to the island of Isla del Coco to study the potential importance of within species behavioral differences and opportunity for allopatry in the process of speciation and niche evolution. Because the focal species *W. galapagensis* shows significant polymorphism in web architecture and because it is found in different microhabitats, the authors argue that it provides a study system that is well fitted to study the diversification processes that take place on small islands.

In order to test if the observed polymorphism in web construction are not indicative for the

presence of otherwise cryptic species, the authors use an genomic approach where ddRAD sequencing is used to generate a large number of SNP for 142 individuals from different locations, microhabitats and with different web types.

The authors generate a large number of SNPs and find no evidence for cryptic species. Thus, they support the idea that this is one highly polymorphic species which has expended its niche significantly in comparison with known congeners.

I find this system very interesting and the result presented here are interesting and largely support the conclusions of the authors.

However, I feel that the ms needs some more work in order to be acceptable for publication in RSPB. Below are my concerns.

Although, the results presented here are interested I find the discussion rather weak as it fails to put the results of this study in a broad context and show how these results specifically help to address the questions mentioned in the introduction and how they help us advance our understanding of the speciation process on islands. A lot of the discussion reviews relevant literature (yet some previous finds that may seem relevant are not discussed) and very few of it is actually discussing the present results.

For example, the authors suggest that invasion of small islands may lead to behavioral polymorphisms, however, there are many examples of striking behavioral polymorphisms in non-island taxa (for example the case of *Philomachus pugnax* is quite similar to the one shown here). Thus, the proposed relationships between behavioral polymorphisms and invasion of small islands shall be substantiated more here. I do not feel that this is more than a speculation given the current level of evidence presented in the ms. If the authors want to keep this point that would warrant some more discussion in my opinion.

I find interesting the evidence that even on a such small island there is a detectable signal of isolation by distance. The authors interpret this as an evidence that the population on the island is not truly panmictic. In such case, and given the rather small numbers of individuals sampled, it would be interesting to get some estimates on population size and some test of assortative mating related to web/habitat types.

Decision letter (RSPB-2020-2045.R0)

25-Sep-2020

Dear Dr Cotoras:

I am writing to inform you that your manuscript RSPB-2020-2045 entitled "Niche partition without speciation: Unparalleled web polymorphism within a single island spider population" has, in its current form, been rejected for publication in Proceedings B.

This action has been taken on the advice of referees, who have recommended that substantial revisions are necessary. With this in mind we would be happy to consider a resubmission, provided the comments of the referees are fully addressed. However please note that this is not a provisional acceptance.

Sincerely,
Dr Locke Rowe
mailto: proceedingsb@royalsociety.org

Associate Editor

Comments to Author:

This study has now been reviewed by two experts in the field and I have read the paper myself. The reviews were mixed, with one Referee recommending Accept with minor revision and the other recommending that major revisions to the study would be needed to be potentially of interest to readers of PRSB. My own take is that elements of the study and system are interesting, but generally speaking the shortcomings identified by both Referees limit the scope and interest of the paper for PRSB. I agree with Referee 1 that it remains unclear how whether the observed behaviours represents intra- or inter-individual plasticity and it is simply difficult to deduce from these restriction site-associated DNA variants, meaning some of the conclusions made cannot be supported by these data and there are outstanding questions about the originality of these findings (both Referees). In addition, the generally limited sample size is seemingly problematic for more informative population genomic inferences about the genetic divergence observed among populations. I agree with both Referees that the discussion tends to exasperate these issues and is both limited in scope while largely speculative. On balance, both Referees provide detailed suggestions on how to potentially overcome these shortcomings, both in terms of revisions and more analyses that could test the central questions being posed. Thus, I would recommend leaving a door open to resubmission that addresses these and the other points raised in these helpful reviews.

Reviewer(s)' Comments to Author:

Referee: 1

Comments to the Author(s)

Review of manuscript RSPB-2020-2045

The present manuscript by Cotoras et al. reports the results from a population genetics study of a species of spider endemic to a small oceanic island that exhibits a remarkable diversity of web morphotypes. By comparing SNP distribution via RADseq between spiders collected from different web phenotypes it was tested, whether the different web morphotypes indicate cryptic species or if this is a case of phenotypic plasticity. The results clearly support the latter hypothesis.

Overall the study appears well conducted – especially the number of samples and loci is quite impressive and probably represents the largest assay of that kind so far for a population of spiders. The writing is very clear, although there are some typos and errors in the language, so the text needs a thorough check. The figures are appropriate.

This is clearly an interesting study and a remarkable example of phenotypic plasticity, but not more.

I am a bit sceptical about the author's discussion and interpretations and strongly suggest a revision. Especially the caveats of the approach must be more clearly communicated.

First, I am not sure if one may speak of 'niche partitioning' without speciation. Maybe the problem is that it remains unclear, what the authors mean by 'niche' in this context. The authors show that there is no correlation between population structure and web morphotype and interpret web diversity as the product of behavioural (phenotypic) plasticity. This means that each individual is capable to produce any of the three web morphotypes depending on the microhabitat structure. The 'niche' concept is usually applied to species, and thus in this case it simply means the species has an enormous niche width, but not that the niche is partitioned in any way between species, not even populations.

Then, it remains unclear if/how different web building behaviour correlates with gene variation and whether the observed phenomenon represents intra- or inter-individual plasticity. It may still be that there are genes correlating with the expression of web building behaviour that were not caught with the chosen RADseq approach. Ideally, the genomic approach would be combined with a behavioural experimental approach, which is easily implemented: Transplanting individuals into different microhabitats and observing the type of web built. For the initial question posed by the authors, it would be very important to clarify this aspect.

Another aspect the authors highlight repeatedly is that this case of web building plasticity is 'outstanding'. But what does that mean. To clarify this the findings should be better embedded into the state of art knowledge on spider web plasticity, as there is a wealth of information on this topic.

So this leaves the question about the originality of these findings. The major finding is that the different web morphotypes are all built by the same species, but this has already been proposed by Bill Eberhard in 1989 ("Niche expansion in the spider *Wendilgarda galapagensis* (Araneae, Theridiosomatidae) on Cocos Island").

Further specific comments below.

Data Statement:

I could not find the Short Read Archive (SRA). Is the NCBI Sequence Read Archive (SRA) meant? An URL should be given.

Title:

I have a problem with the term 'niche partitioning' in this context, as the observed phenomenon clearly is plasticity and no evolutionary process is demonstrated. Also see comment above. I suggest to change or omit the term.

Abstract

l. 39: What is meant with 'population plasticity'? Polymorphism? Also, I would add developmental plasticity here.

l. 45f: This is not a complete sentence. Maybe "This implies that...".

Introduction

l. 57. word is missing here, e.g. "make use of"

l. 122: typo "Galápagos"

l. 136: the radii of an orb web

ll. 140-154: One or three sentences should be added on the function of these web morphotypes, as it is important for the understanding of niche partitioning.

l. 157: typo "a individual"

l. 159: "how often they occur"

l. 162: "web-building" ?

Material and Methods

l. 197: omit "ago" (either the authors refer to the age which is measured in million years or to the origin which is given in million years ago)

Discussion

In the discussion, some aspects are missing. For instance, there is a bulk of information on phenotypic plasticity in spider webs (e.g. reviewed by Boutry, C., & Blamires, S. J. (2013). Plasticity in spider webs and silk: an overview of current evidence. *Spiders: Morphology, Behavior and Geographic Distribution*, 1-46. and Blamires, S. J., Zhang, S., & Tso, I. M. (2017). Webs: diversity, structure and function. In *Behaviour and ecology of spiders*. pp. 137-164. Springer, Cham.)

Also, I would briefly discuss the possibility if the behavioural plasticity may eventually lead to genetic change and speciation according to the theory by West-Eberhard, or if the results rather oppose to that idea.

l. 339. see my comments above on niche and niche partitioning.

l. 344: The other examples given are all much larger species. *Wendilgarda* are extremely small spiders, so it all depends on their dispersal abilities if the island is really 'so small' for them. Is there anything known on the mobility and dispersal abilities of these spiders (e.g. their locomotion radius and if they are able to balloon)?

l. 346f: It is unclear what is meant with this sentence. Needs to be rephrased.

l. 361: So this seems to be an example where environmentally induced developmental plasticity preceded genetic change. This should be discussed.

Figures

Fig. 1: In the web photos it would be good to mark or highlight the sticky lines, so that the reader may better understand the homologous parts and web function. In d, the web structure remains unclear. It looks like not all silk lines were enhanced. At least one additional radius is missing, otherwise the web would collapse. Also, there seem to be additional silk lines underneath, which were not highlighted. Also it is unclear, where the sticky lines are in this picture.

Referee: 2

Comments to the Author(s)

Cotoras and co-authors use a tiny web building spider endemic to the island of Isla del Coco to study the potential importance of within species behavioral differences and opportunity for allopatry in the process of speciation and niche evolution. Because the focal species *W. galapagensis* shows significant polymorphism in web architecture and because it is found in different microhabitats, the authors argue that it provides a study system that is well fitted to study the diversification processes that take place on small islands.

In order to test if the observed polymorphism in web construction are not indicative for the presence of otherwise cryptic species, the authors use a genomic approach where ddRAD sequencing is used to generate a large number of SNPs for 142 individuals from different locations, microhabitats and with different web types.

The authors generate a large number of SNPs and find no evidence for cryptic species. Thus, they support the idea that this is one highly polymorphic species which has expanded its niche significantly in comparison with known congeners.

I find this system very interesting and the result presented here are interesting and largely support the conclusions of the authors.

However, I feel that the ms needs some more work in order to be acceptable for publication in RSPB. Below are my concerns.

Although, the results presented here are interesting I find the discussion rather weak as it fails to put the results of this study in a broad context and show how these results specifically help to address the questions mentioned in the introduction and how they help us advance our understanding of the speciation process on islands. A lot of the discussion reviews relevant

literature (yet some previous finds that may seem relevant are not discussed) and very few of it is actually discussing the present results.

For example, the authors suggest that invasion of small islands may lead to behavioral polymorphisms, however, there are many examples of striking behavioral polymorphisms in non-island taxa (for example the case of *Philomachus pugnax* is quite similar to the one shown here). Thus, the proposed relationships between behavioral polymorphisms and invasion of small islands shall be substantiated more here. I do not feel that this is more than a speculation given the current level of evidence presented in the ms. If the authors want to keep this point that would warrant some more discussion in my opinion.

I find interesting the evidence that even on a such small island there is a detectable signal of isolation by distance. The authors interpret this as an evidence that the population on the island is not truly panmictic. In such case, and given the rather small numbers of individuals sampled, it would be interesting to get some estimates on population size and some test of assortative mating related to web/habitat types.

Author's Response to Decision Letter for (RSPB-2020-2045.R0)

See Appendix A.

RSPB-2020-3138.R0

Review form: Reviewer 1

Recommendation

Accept with minor revision (please list in comments)

Scientific importance: Is the manuscript an original and important contribution to its field?

Good

General interest: Is the paper of sufficient general interest?

Good

Quality of the paper: Is the overall quality of the paper suitable?

Good

Is the length of the paper justified?

Yes

Should the paper be seen by a specialist statistical reviewer?

No

Do you have any concerns about statistical analyses in this paper? If so, please specify them explicitly in your report.

No

It is a condition of publication that authors make their supporting data, code and materials available - either as supplementary material or hosted in an external repository. Please rate, if applicable, the supporting data on the following criteria.

Is it accessible?

Yes

Is it clear?

Yes

Is it adequate?

Yes

Do you have any ethical concerns with this paper?

No

Comments to the Author

After reading the revised ms by Cotoras et al. I find that it has improved greatly compared to its previous version. The translocation experiment is a major addition that takes care of many of the issues with the previous version and clearly increase the relevance of the manuscript. The results of the translocation experiment are indeed very interesting, and they support the conclusions of the authors. However, given that there is no clear population structure and no correlation between any population differentiation and web types, and given that the same individual is able to (and will) build different web types - I have one remaining concern that relates to the use of the term "niche partitioning". I find it hard to see how niche "partitioning" is interpreted here at the level of the individual where there is just high plasticity. Somehow, I feel that "interindividual niche variation", a term that has been used in the literature in similar context, may be more appropriate if one wants to talk about niches here.

In addition, I have couple of suggestions regarding the figures.

Figure 2 - many of the annotations have very small font sizes (e. g. numbers along the axes, axes labels, text in the color legends). Readability will be greatly improved if larger fonts are used.

Figure 3 - it is very hard to see the tips of the arrows.

Decision letter (RSPB-2020-3138.R0)

17-Jan-2021

Dear Dr Cotoras

I am pleased to inform you that your manuscript RSPB-2020-3138 entitled "Intraspecific niche partition without speciation: Individual level web polymorphism within a single island spider population" has been accepted for publication in Proceedings B.

The referee(s) have recommended publication, but also suggest some minor revisions to your manuscript. Therefore, I invite you to respond to the referee(s)' comments and revise your manuscript. Because the schedule for publication is very tight, it is a condition of publication that you submit the revised version of your manuscript within 7 days. If you do not think you will be able to meet this date please let us know.

To revise your manuscript, log into <https://mc.manuscriptcentral.com/prsb> and enter your Author Centre, where you will find your manuscript title listed under "Manuscripts with

Decisions." Under "Actions," click on "Create a Revision." Your manuscript number has been appended to denote a revision. You will be unable to make your revisions on the originally submitted version of the manuscript. Instead, revise your manuscript and upload a new version through your Author Centre.

[http://datadryad.org/submit?journalID=RSPB&manu=\(Document not available\)](http://datadryad.org/submit?journalID=RSPB&manu=(Document%20not%20available)) which will take you to your unique entry in the Dryad repository. If you have already submitted your data to dryad you can make any necessary revisions to your dataset by following the above link. Please see <https://royalsociety.org/journals/ethics-policies/data-sharing-mining/> for more details.

Sincerely,

Dr Locke Rowe

Associate Editor

Board Member

Comments to Author:

This authors have done a thorough job addressing the major comments brought up in the previous version. The addition of the translocation field experiment data was a tremendous addition that went a long way to support the conclusion that individuals can spin more than one web type independently from the web type where they were found and supports the case made using their molecular analyses. The study reads well and the story about these spiders and the extensive field work to realize this experiment will be of interest to readers of PRSB. I tend to agree with the Referee that the concept of “interindividual niche variation” may be more appropriate in the context of niches in this study, but I will leave it up to the authors to decide where they want to settle on the issue. In terms of the translocation data, I have uploaded a marked copy with a few minor revisions about the experimental design that will provide additional clarity for the reader, in addition to several suggested minor edits to improve the clarity of the writing (both in the main text and supplemental). The Referee also makes a few remaining good minor revisions to further improve the manuscript. For all of these reasons I am pleased to recommend Accept with these minor revisions.

Reviewer(s)' Comments to Author:

Referee: 2

Comments to the Author(s).

After reading the revised ms by Cotoras et al. I find that it has improved greatly compared to its previous version. The translocation experiment is a major addition that takes care of many of the issues with the previous version and clearly increase the relevance of the manuscript. The results of the translocation experiment are indeed very interesting, and they support the conclusions of the authors. However, given that there is no clear population structure and no correlation between any population differentiation and web types, and given that the same individual is able to (and will) build different web types - I have one remaining concern that relates to the use of the term “niche partitioning”. I find it hard to see how niche “partitioning” is interpreted here at the level of the individual where there is just high plasticity. Somehow, I feel that “interindividual niche variation”, a term that has been used in the literature in similar context, may be more appropriate if one wants to talk about niches here.

In addition, I have couple of suggestions regarding the figures.

Figure 2 - many of the annotations have very small font sizes (e. g. numbers along the axes, axes labels, text in the color legends). Readability will be greatly improved if larger fonts are used.

Figure 3 - it is very hard to see the tips of the arrows.

Author's Response to Decision Letter for (RSPB-2020-3138.R0)

See Appendix B.

Decision letter (RSPB-2020-3138.R1)

22-Jan-2021

Dear Dr Cotoras

I am pleased to inform you that your manuscript entitled "Intraspecific niche partition without speciation: Individual level web polymorphism within a single island spider population" has been accepted for publication in Proceedings B.

Open Access

Paper charges

You are allowed to post any version of your manuscript on a personal website, repository or preprint server. However, the work remains under media embargo and you should not discuss it

with the press until the date of publication. Please visit <https://royalsociety.org/journals/ethics-policies/media-embargo> for more information.

Sincerely,
Proceedings B
<mailto:proceedingsb@royalsociety.org>

Appendix A

Associate Editor

Comments to Author:

This study has now been reviewed by two experts in the field and I have read the paper myself. The reviews were mixed, with one Referee recommending Accept with minor revision and the other recommending that major revisions to the study would be needed to be potentially of interest to readers of PRSB. My own take is that elements of the study and system are interesting, but generally speaking the shortcomings identified by both Referees limit the scope and interest of the paper for PRSB. I agree with Referee 1 that it remains unclear how whether the observed behaviours represents intra- or inter-individual plasticity and it is simply difficult to deduce from these restriction site-associated DNA variants, meaning some of the conclusions made cannot be supported by these data and there are outstanding questions about the originality of these findings (both Referees). In addition, the generally limited sample size is seemingly problematic for more informative population genomic inferences about the genetic divergence observed among populations. I agree with both Referees that the discussion tends to exasperate these issues and is both limited in scope while largely speculative. On balance, both Referees provide detailed suggestions on how to potentially overcome these shortcomings, both in terms of revisions and more analyses that could test the central questions being posed. Thus, I would recommend leaving a door open to resubmission that addresses these and the other points raised in these helpful reviews.

Cotoras et al: Thank you very much for the editorial work and critically assessing the reviewers comments. We are happy to hear that you find our work interesting and appreciate that you highlight the most critical issues to address. In the following paragraphs we have replied point-by-point the different comments and suggestions.

In particular, we would like to mention that we added a data set from a translocation field experiment, which directly tests the capability of an individual to spin more than one web type. The results parallel the genomic data showing that individuals, independently from the web type where they were found, can spin other web types. Interestingly, the transitions between web types are not all equal and there are clear preferences. We discuss these findings in the light of the genomic data and in the context of niche expansion of this insular species. The field translocation experiment was originally intended to be on a separate paper, but based on the reviewer's suggestion we decided to incorporate it here, and we agree that it greatly supports the case made using molecular markers.

Regarding the originality of our findings, while they definitely built on the work of Eberhard (1989), we provide 2 key pieces of novel information. First, we show conclusive proof using cutting edge high throughput sequencing that *W. galapagensis* corresponds to a single species without geographic structure. Before our work, the only evidence to support this idea was Archer's species description (1953), which is very succinct and based only on museum specimens. Archer did not visit the island or reported the web variation, indeed the name of the species reflects his mistake of considering that it was collected in the Galápagos archipelago. The second piece of novel information is the field experiment of microhabitat translocation. These observations demonstrate the plastic individual behavior and the existence of clear preferences (not all transitions are equally likely). This result corresponds to a systematic approach to the incidental observations (without individual marking) made by professor William Eberhard about two out of five individuals observed to change from Land web to Water web.

This observation was taken by us as preliminary data for our experiment and as evidence for our single population hypothesis. But, our experiment is the one that in a systematic way confirms the plasticity and adds the dimension of unequal transition probabilities between web types.

We have revisited the extent of our conclusions in order to strictly stick to what the genomic and behavioral data allows. In addition, many previously speculative statements now are supported by the data coming from the translocation field experiment.

We hope these additions, improvements and clarifications have risen the quality of our work to the level of acceptance for publication at PRSB. Thank you very much for your contributions to the improvement of our research.

Reviewer(s)' Comments to Author:

Referee: 1

Comments to the Author(s)

Review of manuscript RSPB-2020-2045

The present manuscript by Cotoras et al. reports the results from a population genetics study of a species of spider endemic to a small oceanic island that exhibits a remarkable diversity of web morphotypes. By comparing SNP distribution via RADseq between spiders collected from different web phenotypes it was tested, whether the different web morphotypes indicate cryptic species or if this is a case of phenotypic plasticity. The results clearly support the latter hypothesis.

Overall the study appears well conducted – especially the number of samples and loci is quite impressive and probably represents the largest assay of that kind so far for a population of spiders. The writing is very clear, although there are some typos and errors in the language, so the text needs a thorough check. The figures are appropriate. This is clearly an interesting study and a remarkable example of phenotypic plasticity, but not more.

Cotoras et al: Thank you very much for your review work and appreciating our research. We are happy to hear the positive evaluation on the overall implementation, sampling effort, writing and figures of our study. Along the same line, we are glad to see that the reviewer find our study system interesting.

On the following paragraphs, we will present point by point the improvements on the manuscript based on the suggestions. Thank you very much for the critic and constructive comments.

I am a bit sceptical about the author's discussion and interpretations and strongly suggest a revision. Especially the caveats of the approach must be more clearly communicated. First, I am not sure if one may speak of 'niche partitioning' without speciation. Maybe the problem is that it remains unclear, what the authors mean by 'niche' in this context.

Cotoras et al: The meaning of 'niche' that we apply is based on MacArthur (1958): "... species can coexist only if each inhibits its own population more than the others'. This is probably equivalent to saying that species divide up the resources of a community in such a way that each

species is limited by a different factor. (...) Coexistence in one habitat, then, may be the result of each species being limited by the availability of a resource in different habitats." In other words, niche partition takes place when different organisms use of ecological space in their own ways. In particular, we consider that individuals presenting the different web types are occupying different niches related to microhabitat and trophic interactions. Each web type is located in a different microhabitat, which exposes the spider to different environmental conditions (wind, humidity, exposure to predators, possibility of being destroyed by overflow, etc.). Related with that, each microhabitat provides a different set of potential preys. The spiders with a Water web could capture insects hovering or walking over water, while the ones with Aerial webs are restricted to winged insects. In the case of the Land web, their most likely prey are insects walking on dry land. Therefore, these differences related to environmental conditions and prey availability are the conceptual base to refer to different niches. In other words, the different web building behaviors result in different niche constructions itself (Blamires, 2013).

As for the restriction of the niche concept to only a species-level property, please see the following reply.

The authors show that there is no correlation between population structure and web morphotype and interpret web diversity as the product of behavioural (phenotypic) plasticity. This means that each individual is capable to produce any of the three web morphotypes depending on the microhabitat structure. The 'niche' concept is usually applied to species, and thus in this case it simply means the species has an enormous niche width, but not that the niche is partitioned in any way between species, not even populations.

Cotoras et al: The reviewer is correct on the observation that the concept of 'niche' is usually applied to species. However, the concept of intraspecific niche partition is present in the literature and discussed in the context of generalist species (Cloyd & Eason, 2017), species with large phenotypic variation (Benard & Maher, 2011) and comprehensively reviewed for more than 90 species distributed across many taxonomic groups (Bolnick et al., 2003).

Cloyd & Eason (2017) mentioned that "This intraspecific niche variation caused by body size [the study measures this specific trait] can potentially alter competitive interactions and affect niche partitioning (...) Intraspecific niche variation associated with individual size may result from both ontogenetic change and individual variation.". Congruently, the title of the study by Benard & Maher (2011) reads: "Consequences of intraspecific niche variation: phenotypic similarity increases competition among recently metamorphosed frogs".

The review by Bolnick et al. (2003) summarizes these ideas and place them on a broad context: "Most empirical and theoretical studies of resource use and population dynamics treat conspecific individuals as ecologically equivalent. This simplification is only justified if interindividual niche variation is rare, weak, or has a trivial effect on ecological processes. (...) The degree of individual specialization varies widely among species and among populations, reflecting a diverse array of physiological, behavioral, and ecological mechanisms that can generate intrapopulation variation. Finally, individual specialization has potentially important ecological, evolutionary, and conservation implications. Theory suggests that niche variation facilitates frequency-dependent interactions that can profoundly affect the population's stability, the amount of intraspecific competition, fitness-function shapes, and the population's capacity to diversify and speciate rapidly."

Indeed, the concept of intraspecific niche partition is also mentioned as a potential explanation for the presence of nine different foraging behaviours on the Darwin's finch from Isla del Coco (Werner & Sherry, 1987): "We suggest that the foraging specializations of individual Cocos Finches could provide a powerful within-population model for niche differentiation by species within larger faunas. This analogy would be particularly valid if intraspecific competition is prerequisite to intrapopulation specializations in the same way that interspecific competition is thought to be an integral component of species divergence in the course of adaptive radiation."

The acknowledgement of the classic correlation between one niche and one species, was one of the justifications for our hypothesis about the presence of really three cryptic species, each with a different and well-defined niche. In the second paragraph of the introduction, we mention that "Initial stages of niche expansion, prior to niche partition, might require the evolution of phenotypic polymorphism...", so the reviewer is correct on arguing that this might well be described as a species with a wide niche, instead of a species with intraspecific niche partition (Werner & Sherry, 1987; Bolnick et al., 2003; Benard & Maher, 2011; Cloyed & Eason, 2017). However, we refer to niche partition based on the results of translocation experiments previously not included in the manuscript. These experiments (see improved version for details) show that even the spiders present some degree of plasticity at the individual level, they do present preferences which restrict some individuals to have a strong tendency to use one kind of web instead of the other.

One of the main take home messages from our work is that in adaptive radiations common in large and complex archipelagoes systems, the radiating clade expands its niche and each individual species occupies a fraction of it. While, in intraspecific adaptive radiations, postulated to be more common on small islands, it is a single species the one which expands its niche and individuals dynamically will occupy different sections of this wider niche.

Cloyed, C.S. & Eason, P.K. (2017) Niche partitioning and the role of intraspecific niche variation in structuring a guild of generalist anurans. *Royal Society Open Science*, **4**, 170060.

Benard, M.F. & Maher, J.M. (2011) Consequences of intraspecific niche variation: phenotypic similarity increases competition among recently metamorphosed frogs. *Oecologia*, **166**, 585-592.

Bolnick, D.I., Svanbäck, R., Fordyce, J.A., Yang, L.H., Davis, J.M., Hulsey, C.D. & Forister, M.L. (2003) The ecology of individuals: incidence and implications of individual specialization. *The American Naturalist*, **161**, 1-28.

Werner, T.K. & Sherry, T.W. (1987) Behavioral feeding specialization in *Pinaroloxias inornata*, the "Darwin's Finch" of Cocos Island, Costa Rica. *Proceedings of the National Academy of Sciences of the United States of America*, **84**, 5506-5510.

Then, it remains unclear if/how different web building behaviour correlates with gene variation and whether the observed phenomenon represents intra- or inter-individual plasticity. It may still be that there are genes correlating with the expression of web building behaviour that were not caught with the chosen RADseq approach.

Cotoras et al: The main objective of our genomic approach was not to detect genes related with different web building behavior. Instead, we were interested in capturing independent markers to

test if there was differential population grouping defined by web type. This signal of differential grouping would be given by the disruption of gene flow in the case of cryptic species.

It is true that genes specifically related with web building behavior might not have been selected with the ddRAD approach, but in any case it was not our main interest. We did search for F_{st} outliers, which represent genes under strong selection. The hypothesis was that those loci might be part of genes involved on a trait under positive selection (i.e. web constriction). However, we did not find differences between populations of different web types based on those outliers. Therefore, there is no signature of a genetic trait which is separating individuals collected from different web types. These results are reinforced by analysis of field translocation experiments included in this manuscript version, as the reviewer suggested.

Ideally, the genomic approach would be combined with a behavioural experimental approach, which is easily implemented: Transplanting individuals into different microhabitats and observing the type of web built. For the initial question posed by the authors, it would be very important to clarify this aspect.

Cotoras et al: Thank you very much for pointing this out. We absolutely agree with the reviewer on this point and indeed we performed exactly the suggested experiment in two later visits to the island. This data set, because it was created after the genomic study, was originally intended to be published on a separated paper. But, we agree that these data strengthen our conclusions and have included them here.

The results from translocation experiments are congruent with the genomic observations. We were able to demonstrate that the same individual is able to spin more than one web type. This situation was documented on individuals coming from different web types. In addition, we show that there are clear preferences on the types of webs that spiders will spin after an experimental perturbation (translocation or control).

Another aspect the authors highlight repeatedly is that this case of web building plasticity is ‘outstanding’. But what does that mean. To clarify this the findings should be better embedded into the state of art knowledge on spider web plasticity, as there is a wealth of information on this topic.

Cotoras et al: Following the reviewer’s suggestion, in the Introduction on the sixth paragraph of the sub-section “*The unique web polymorphism of Wendilgarda galapagensis Archer, 1953*” it is possible to find an explanation and references to justify the classification as ‘outstanding’ to the web plasticity.

Here, we mention how the differences between the three web types of *W. galapagensis* go beyond changes in the number of repetitions of a given architectural element or relative sizes. They differ in the general structure, microhabitat placement and behavior associated to its constructions. Considering the high degree of evolutionary conservatism of the web architecture as a trait, which in many cases could be used as a family-level diagnostic character, we argue that the plasticity in *W. galapagensis* is exceptional. It is comparable to variation in morphospace occupied by the cranial shape of Hawaiian Honey creepers, which is one of the highest reported among bird families (Tokita et al, 2016),

Tokita M, Yano W, James HF and Abzhanov A. 2017. Cranial shape evolution in adaptive radiations of birds: comparative morphometrics of Darwin's finches and Hawaiian honeycreepers. *Phil. Trans. R. Soc. B* 372: 20150481.

So this leaves the question about the originality of these findings. The major finding is that the different web morphotypes are all built by the same species, but this has already been proposed by Bill Eberhard in 1989 (“Niche expansion in the spider *Wendilgarda galapagensis* (Araneae, Theridiosomatidae) on Cocos Island”).

Cotoras et al: Our findings definitely built on the work of Eberhard (1989), but they provide 2 key pieces of novel information. First, we provide conclusive proof using cutting edge high throughput sequencing that *W. galapagensis* corresponds to a single species without geographic structure. Before our work, the only evidence to support this idea was Archer's species description (1953), which is very succinct and based only on museum specimens. Archer did not visit the island or reported the web variation, indeed the name of the species reflects his mistake of considering that it was collected in the Galápagos archipelago. The second piece of novel information is the field experiment of microhabitat translocation. These observations demonstrate the plastic individual behavior and the existence of clear preferences (not all transitions are equally likely). This result corresponds to a systematic approach to the incidental observations made by professor William Eberhard (1989) about two out of five individuals observed to change from Land web to Water web. This observation was taken by us as preliminary data for our experiment and as evidence for our single population hypothesis, but it was conducted without individual marking, so the identity of spiders with different web types could not be ascertained. Our experiment is the one that in a systematic way confirms the plasticity and adds the dimension of unequal transition probabilities between web types.

Further specific comments below.

Data Statement:

I could not find the Short Read Archive (SRA). Is the NCBI Sequence Read Archive (SRA) meant? An URL should be given.

Cotoras et al: Our apologies for the typo. Yes, SRA is the NCBI Sequence Read Archive. We have corrected the word and added the URL.

Title:

I have a problem with the term ‘niche partitioning’ in this context, as the observed phenomenon clearly is plasticity and no evolutionary process is demonstrated. Also see comment above. I suggest to change or omit the term.

Cotoras et al: Thank you very much for the observations about the use of “niche partitioning”. They have been useful to better explain our original ideas and we have made numerous changes to clarify our thought process.

We have modified the beginning of title by “Intraspecific niche partitioning...”, instead of “Niche partitioning...”. We have added the term “Intraspecific” to distinguish from the classic use of “niche partition” in the context of adaptive radiations. On adaptive radiations, each

partition of the niche is used exclusively by a single species, usually characterized by an eco-morphology. But, in the case of *W. galapagensis*, the different niches -characterized by web types- are used by individuals within the same species. As shown on the newly added transplantation experiment, individuals are able to switch from one web type to the other.

In other words, the niche partition accomplished by the different web types of *W. galapagensis* is intraspecific and dynamic in the sense of not been permanently fix as what is observed between species from different eco-morphologies on an adaptive radiation. On the third reply we provide a more expense discussion on the concept of intraspecific niche partition.

Abstract

I. 39: What is meant with ‘population plasticity’? Polymorphism? Also, I would add developmental plasticity here.

Cotoras et al: Yes, by ‘population plasticity’ we refer to population polymorphism. We have added the concept of developmental plasticity.

I. 45f: This is not a complete sentence. Maybe “This implies that...”.

Cotoras et al: Corrected

Introduction

I. 57. word is missing here, e.g. “make use of”

Cotoras et al: Corrected

I. 122: typo “Galápagos”

Cotoras et al: Corrected

I. 136: the radii of an orb web

Cotoras et al: Added

II. 140-154: One or three sentences should be added on the function of these web morphotypes, as it is important for the understanding of niche partitioning.

Cotoras et al: This is a very good suggestion. Thank you very much. We have added a paragraph where we explain what we understand by different niches, in particular referring to the microhabitats and trophic interactions (prey availability) that each web type is exposed to.

I. 157: typo “a individual”

Cotoras et al: Corrected

I. 159: “how often they occur”

Cotoras et al: Corrected

l. 162: “web-building” ?

Cotoras et al: Corrected

Material and Methods

l. 197: omit “ago” (either the authors refer to the age which is measured in million years or to the origin which is given in million years ago)

Cotoras et al: Corrected. Thank you very much for the clarification as to when is appropriate to use “ago”.

Discussion

In the discussion, some aspects are missing. For instance, there is a bulk of information on phenotypic plasticity in spider webs (e.g. reviewed by Boutry, C., & Blamires, S. J. (2013). *Plasticity in spider webs and silk: an overview of current evidence. Spiders: Morphology, Behavior and Geographic Distribution*, 1-46. and Blamires, S. J., Zhang, S., & Tso, I. M. (2017). *Webs: diversity, structure and function. In Behaviour and ecology of spiders*. pp. 137-164. Springer, Cham.)

Cotoras et al: Thank you very much for the literature suggestion. Both reviews were very insightful on ideas and directed us to additional relevant literature.

Now, in the Introduction, we have placed the case of *W. galapagensis* in the context of other species (orb webs and combwebs, reviewed by Boutry & Blamires, 2013) with architectural web variation. Indeed, the fact that many spiders do present plasticity on their web structures it is presented as an argument to support the hypothesis that what is observed in *W. galapagensis* (3 different web types) corresponds to plasticity and it is not the indication of different cryptic species.

Moreover in the Discussion, we have added a paragraph explaining how the argument of competitive exclusion as a driver for web plasticity has elements from previously described modes to explain plasticity on web architecture. In particular, from models about Optimal Foraging, Optimal Performance and Plastic Extended Phenotype (reviewed by Blamires et al, 2017).

Also, I would briefly discuss the possibility if the behavioural plasticity may eventually lead to genetic change and speciation according to the theory by West-Eberhard, or if the results rather oppose to that idea.

Cotoras et al: Thank you very much for this observation. Indeed, it refers to the major take home messages of our manuscript.

The last paragraphs of the Discussion have been improved to reflect this point and link with the idea of niche partitions. In summary, we say that on an adaptive radiation the presence of allopatric barriers allows for speciation and the later occupation of different niches by the new species in order to reduce inter-specific competition. But, in cases of “intraspecific adaptive radiation” (sensu West-Eberhard, 2003) the lack of allopatric barriers will prevent speciation; yet

in order to reduce intra-specific intraspecific, competition an expanded species niche will arise. Dynamically different fractions of this expanded niche will be occupied by individuals, with different web types on this case.

In other words, in contrast with the species specific well-defined niche partitions characteristic of adaptive radiations; on an intraspecific adaptive radiation (associated with a single species) individuals are able to use different fractions of the total species niche in different moments.

l. 339. see my comments above on niche and niche partitioning.

Cotoras et al: Accordingly to the previously suggested modifications, we have added an explanation indicating that the niche partitioning occurring in *W. galapagensis* is intraspecific and dynamic. While, it resembles the niche partitions existing between eco-morphologies of an adaptive radiation, it is not fixed and allows for the same individual to use different web types on different moments.

l. 344: The other examples given are all much larger species. *Wendilgarda* are extremely small spiders, so it all depends on their dispersal abilities if the island is really 'so small' for them. Is there anything known on the mobility and dispersal abilities of these spiders (e.g. their locomotion radius and if they are able to balloon)?

Cotoras et al: It is correct that comparably *Wendilgarda* is smaller in size than *Tetragnatha* or *Dysdera*. It could be considered as a proxy to predict a reduced mobility. This reduced mobility could ultimately result in that even a small islands, such as Isla del Coco, might be "large enough" for the species to find barriers to gene flow.

There are not studies on locomotion radius or ballooning in *W. galapagensis* or other *Wendilgarda* species. But, our population genetic study demonstrates that while there is some isolation by distance, it is weak, with populations being weakly differentiated ($F_{st} \sim 0.009$). Furthermore, web architecture is not structured by population. The signal of isolation by distance therefore does not leave a signature of overall population structure. Therefore, we can argue that the island is in fact small enough for the species to not differentiate populations.

l. 346f: It is unclear what is meant with this sentence. Needs to be rephrased.

Cotoras et al: Corrected. The intention of the phrase was to talk about the islands, not the spiders. It has been re-written to express the correct idea.

l. 361: So this seems to be an example where environmentally induced developmental plasticity preceded genetic change. This should be discussed.

Cotoras et al: That is correct. At the end of the paragraph, we relate this example with *W. galapagensis*. In particular, referring to the fact that in contrast to the *Howea* palm, on *W. galapagensis* it is not clear what kind of barrier for gene flow could be acting.

Figures

Fig. 1: In the web photos it would be good to mark or highlight the sticky lines, so that the

reader may better understand the homologous parts and web function. In d, the web structure remains unclear. It looks like not all silk lines were enhanced. At least one additional radius is missing, otherwise the web would collapse. Also, there seem to be additional silk lines underneath, which were not highlighted. Also it is unclear, where the sticky lines are in this picture.

Cotoras et al: Thank you very much for the suggestions. We have added arrows and arrow heads to indicate the sticky lines and horizontal suspension lines, respectively on (b) and (c).

We agree with the reviewer that one silk line is not enhanced on (d). But, that is because the picture did not capture the missing line (probably the lower right corner). See Supp. File 1 for the original picture. Related to mentioned additional silk line present at the bottom of the picture, it was not enhanced because it appears to be from another web.

The sticky lines in the Aerial Webs radiate from the central area of the web and tend to be fewer and longer than those from other two types of webs (Eberhard, 1989) (information now added to the description of the web types on the Introduction). But, due to this similarity with horizontal lines, they cannot be distinguished easily from a picture.

Referee: 2

Comments to the Author(s)

Cotoras and co-authors use a tiny web building spider endemic to the island of Isla del Coco to study the potential importance of within species behavioral differences and opportunity for allopatry in the process of speciation and niche evolution. Because the focal species *W. galapagensis* shows significant polymorphism in web architecture and because it is found in different microhabitats, the authors argue that it provides a study system that is well fitted to study the diversification processes that take place on small islands.

In order to test if the observed polymorphism in web construction are not indicative for the presence of otherwise cryptic species, the authors use a genomic approach where ddRAD sequencing is used to generate a large number of SNPs for 142 individuals from different locations, microhabitats and with different web types.

The authors generate a large number of SNPs and find no evidence for cryptic species.

Thus, they support the idea that this is one highly polymorphic species which has expanded its niche significantly in comparison with known congeners.

I find this system very interesting and the result presented here are interesting and largely support the conclusions of the authors.

However, I feel that the ms needs some more work in order to be acceptable for publication in RSPB. Below are my concerns.

Cotoras et al: Thank you very much for your review work and explicitly describing the main take home messages from our research. We are very happy to hear that you found our study system very interesting and agree with the conclusions we have drawn from our results.

Thank you for your constructive suggestions to make our work acceptable for publication in RSPB. Please see below and in the text the implementation of all the improvements.

Although, the results presented here are interesting I find the discussion rather weak as it fails to put the results of this study in a broad context and show how these results

specifically help to address the questions mentioned in the introduction and how they help us advance our understanding of the speciation process on islands. A lot of the discussion reviews relevant literature (yet some previous finds that may seem relevant are not discussed) and very few of it is actually discussing the present results.

For example, the authors suggest that invasion of small islands may lead to behavioral polymorphisms, however, there are many examples of striking behavioral polymorphisms in non-island taxa (for example the case of *Philomachus pugnax* is quite similar to the one shown here). Thus, the proposed relationships between behavioral polymorphisms and invasion of small islands shall be substantiated more here. I do not feel that this is more than a speculation given the current level of evidence presented in the ms. If the authors want to keep this point that would warrant some more discussion in my opinion.

Cotoras et al: Thank you for your suggestions to make our research more relevant in a broad context. Following them, we have improved our discussion in the following points:

1.- Added a whole new data set corresponding to a microhabitat translocation experiments in order to demonstrate the ability of an individual to spin more than one web type. Originally, this evidence was going to be presented on a second paper, but based on Reviewer #1's suggestion we incorporated it here. The results are congruent with our genomic data showing that a single individual is indeed able to spin more than one web type. Interestingly, the transition between web types is not equal in all cases, demonstrating preferences. All of this is currently incorporated into the Discussion and integrated with genomic data.

2.- Further developed the idea that individuals are able to dynamically use different fractions of the expanded niche of this insular species.

3.- Interpreted the web architectural plasticity under the light of previously proposed models (reviewed by Boutry and Blamires, 2013).

4.- Explicitly indicate that niche expansion followed by island colonization is due to ecological release (Kohn, 1972; 1978).

5.- Further discussing the differences regarding niche occupation between a multi-species adaptive radiation from a large archipelago system, and an intraspecific adaptive radiation from a small island.

6.- Presenting and discussing additional examples where the pattern of intraspecific adaptive radiation can be recognized (*Conus miliaris pascuensis* and *Conus californicus*).

I find interesting the evidence that even on a such small island there is a detectable signal of isolation by distance. The authors interpret this as an evidence that the population on the island is not truly panmictic. In such case, and given the rather small numbers of individuals sampled, it would be interesting to get some estimates on population size and some test of assortative mating related to web/habitat types.

Cotoras et al: We appreciate the reviewer's suggestion, but a population size estimation does not relate directly with the answer of our original question. Moreover, considering all the assumptions that go into it and the lack of basic knowledge of the species (for example, generation time), we prefer to not include such calculation as it will likely be highly inaccurate. Related to the assortative mating idea, we do not consider necessary to perform that test, as it is clear from the PCA analysis and F_{st} that there is no grouping by web type or locality (Fig. 4). It is important to remember that isolation-by-distance merely shows that genetic and geographic

distances are correlated, which is typical of most species. The populations could be near-panmictic and still show signs of geographic structuring. This is indeed what happens with this spider, as the populations themselves are very weakly differentiated ($F_{st} \sim 0.009$). The genetic clustering by web type was also explored on an individual site level with the same result (Supp. Fig. 3).

Appendix B

Associate Editor

Board Member

Comments to Author:

This authors have done a thorough job addressing the major comments brought up in the previous version. The addition of the translocation field experiment data was a tremendous addition that went a long way to support the conclusion that individuals can spin more than one web type independently from the web type where they were found and supports the case made using their molecular analyses. The study reads well and the story about these spiders and the extensive field work to realize this experiment will be of interest to readers of PRSB.

Cotoras et al: We are very happy to hear that our additions and replies to the reviewer's comments have successfully addressed the previous concerns on our manuscript. We are also grateful for their evaluation work, which has guided this improvement.

I tend to agree with the Referee that the concept of “interindividual niche variation” may be more appropriate in the context of niches in this study, but I will leave it up to the authors to decide where they want to settle on the issue.

Cotoras et al: Thank you for allowing us to decide on the terminology use to describe this phenomenon. We have finally decided to keep the concept of “intraspecific niche partition”, because it better describes the situation where each individual can only occupy a section of the species niche at the given time.

In terms of the translocation data, I have uploaded a marked copy with a few minor revisions about the experimental design that will provide additional clarity for the reader, in addition to several suggested minor edits to improve the clarity of the writing (both in the main text and supplemental).

Cotoras et al: We very much appreciate the editor taking the time to directly add to the texts edits on the writing and comments on the experimental design. We have incorporated them on the revised version.

The Referee also makes a few remaining good minor revisions to further improve the manuscript.

Cotoras et al: They have been incorporated.

For all of these reasons I am pleased to recommend Accept with these minor revisions.

Reviewer(s)' Comments to Author:

Referee: 2

Comments to the Author(s).

After reading the revised ms by Cotoras et al. I find that it has improved greatly compared

to its previous version. The translocation experiment is a major addition that takes care of many of the issues with the previous version and clearly increase the relevance of the manuscript. The results of the translocation experiment are indeed very interesting, and they support the conclusions of the authors.

Cotoras et al: We are very happy to hear that the addition of the experimental data set supports and improves the clarity of our main message.

However, given that there is no clear population structure and no correlation between any population differentiation and web types, and given that the same individual is able to (and will) build different web types - I have one remaining concern that relates to the use of the term “niche partitioning”. I find it hard to see how niche “partitioning” is interpreted here at the level of the individual where there is just high plasticity. Somehow, I feel that “interindividual niche variation”, a term that has been used in the literature in similar context, may be more appropriate if one wants to talk about niches here.

Cotoras et al: Thank you very much for bring up to our attention the concept of “interindividual niche variation”. We now mention the use of this term on similar biological contexts (Costa et al., 2015; Costa-Pereira et al., 2018) on Line 480.

Our main reason to prefer the use of “niche partitioning” is the fact that it is possible to identify discrete strategies associated to the niche occupation. In the case of *W. galapagensis* those discrete strategies correspond to the different web types and their respective microhabitat. It is true that these different strategies represent interindividual niche variation. But, as each individual can exclusively occupy a single section of the species niche at a given time (despite they can switch over time), we think “niche partition” is a better description for the phenomenon.

Costa, A., Salvidio, S., Posillico, M., Matteucci, G., De Cinti, B., & Romano, A. (2015) Generalisation within specialization: inter-individual diet variation in the only specialized salamander in the world. *Scientific Reports*, **5**, 13260.

Costa-Pereira, R., Rudolf, V.H.W., Souza, F.L., & Araújo, M.S. (2018) Drivers of individual niche variation in coexisting species. *Journal of Animal Ecology*, **87**, 1-13.

In addition, I have couple of suggestions regarding the figures.

Figure 2 - many of the annotations have very small font sizes (e. g. numbers along the axes, axes labels, text in the color legends). Readability will be greatly improved if larger fonts are used.

Cotoras et al: Thank you very much for the suggestion. Now the annotations have a larger font size.

Figure 3 - it is very hard to see the tips of the arrows.

Cotoras et al: Thank you for the observation. We have increased the size and thickness of the arrows in the figure.